# Transparent near-infrared perovskite light-emitting diodes

Chenchao Xie[1,2], Xiaofei Zhao[1], Evon Woan Yuann Ong[1,2] & Zhi-Kuang Tan [1,2✉]

Mobile and wearable devices are increasingly reliant on near-infrared (NIR) covert illumination for facial recognition, eye-tracking or motion and depth sensing functions. However, these small devices offer limited spatial real estate that is typically already occupied by their full-area electronic color displays. Here, we report a transparent perovskite light-emitting diode (LED) that could be overlaid across a color display to provide an efficient and high-intensity NIR illumination. Our transparent devices are constructed with an ITO/AZO/PEIE/FAPbI$_3$/poly-TPD/MoO$_3$/Al/ITO/Ag/ITO architecture, and offer a high average transmittance of more than 55% across the visible spectral region. In particular, our Al/ITO/Ag/ITO top transparent electrode was designed to offer a combination low sheet resistance and low plasma damage upon electrode deposition. The devices emit at 799 nm with a high total external quantum efficiency of 5.7% at a current density of 5.3 mA cm$^{-2}$ and a high radiance of 1.5 W sr$^{-1}$ m$^{-2}$, and possess a large functional device area of 120 mm$^2$. The efficient performance is ideal for battery-powered wearable devices, and could enable advanced security and sensing features on future smart-watches, phones, gaming consoles and augmented or virtual reality headsets.

[1] Department of Chemistry, National University of Singapore, 3 Science Drive 3, 117543 Singapore, Singapore. [2] Solar Energy Research Institute of Singapore, National University of Singapore, 7 Engineering Drive 1, 117574 Singapore, Singapore. ✉email: chmtanz@nus.edu.sg

Perovskite light emission research has advanced rapidly, with the efficiencies of electroluminescent devices progressing quickly from 0.8% in earlier works[1–5] to over 20% in more recent reports[6–11]. NIR perovskite LEDs (PeLEDs) have remained the most successful, demonstrating a combination of high efficiency, good reproducibility, and respectable lifespan. Recently, efficient large-area devices have also been realized, and were demonstrated to possess remarkable emission uniformity in both rigid and flexible form factors[11].

The primary advantage that PeLEDs offer over other III-V semiconductor chip-based LEDs lies in their ability to be constructed over large areas on a variety of substrates, thus allowing them to be suited for electroluminescent display applications such as in televisions, smartphones, and smart-watches. These are similar to the value propositions offered by organic light-emitting diodes and quantum dot LEDs. However, a NIR LED, which has conventionally been associated with covert illumination, security, and optical communication functions, would presently serve limited functions if produced over a large or flexible area.

Here, we report the realization of a transparent, large-area NIR-emitting PeLED and demonstrate, as a proof-of-concept, its unique functional applications when overlaid across an emissive color display. Such transparent NIR devices would offer an array of advanced security and sensing functions on tech-gadgets without occupying additional real estate, potentially even freeing up space for extra functionalities. Examples of advanced features could include facial recognition, eye-tracking or motion and depth sensing on smart-watches, phones, gaming consoles, and augmented reality or virtual reality devices.

## Results

**Transparent device characteristics.** Our transparent PeLEDs are constructed with an indium tin oxide (ITO)/aluminum zinc oxide (AZO)/polyethylenimine ethoxylated (PEIE)/formamidinium lead iodide (FAPbI$_3$)/poly[N,N′-bis(4-butylphenyl)-N,N′-bisphenylbenzidine] (poly-TPD)/molybdenum trioxide (MoO$_3$)/aluminum (Al)/ITO/ silver (Ag)/ITO architecture (Fig. 1a), and employ similar active layers as the opaque devices in our previous work[11]. The scanning electron microscopy image and the X-ray diffraction pattern of the FAPbI$_3$ perovskite layer are shown in Supplementary Figs. 1 and 2, respectively. Notably in this work, the top electrode has been designed with a multilayered Al (10 nm)/ITO (40 nm)/Ag (10 nm)/ITO (40 nm) structure to possess a combination of low sheet resistance, efficient charge injection, and high optical transparency. The MoO$_3$/poly-TPD and AZO/PEIE layers were employed to facilitate ohmic and balanced injection of holes and electrons into the perovskite, respectively, for efficient electroluminescence (EL).

Figure 1b shows the EL spectra of the transparent PeLED, with characteristic NIR emission at 799 nm, which is consistent with the photoluminescence spectrum of FAPbI$_3$ (Supplementary Fig. 3). Our devices were fabricated with a large area of 120 mm$^2$ (15 × 8 mm), and show remarkably uniform emission across the entire active area. Figure 1c shows the combined current density vs. voltage and radiance vs. voltage plots of the transparent PeLED. The device turns on at a low voltage of ~1.5 V, thus indicating efficient carrier injection from both electrodes. Since our device contains transparent electrodes on both sides, we define the emission from the substrate side as front emission and that from the Al/ITO/Ag/ITO side as back emission. The front emission is more intense and reaches a maximum radiance of 2.8 W sr$^{-1}$ m$^{-2}$ and while the back emission has a radiance of 1.2 W sr$^{-1}$ m$^{-2}$ at a driving voltage of 4.0 V. This is due to the higher transmittance of the front ITO-glass substrate (Supplementary Fig. 4) as well as contributions of reflection from the thin metallic interlayers in the back electrode.

As a consequence, the external quantum efficiency (EQE) (Fig. 1d) is calculated to be 4.5% and 1.2% for the front and back emission, respectively, hence giving a total EQE of 5.7% at a current density of 5.3 mA cm$^{-2}$ and a corresponding total radiance of 1.5 W sr$^{-1}$ m$^{-2}$. The average max-EQE for the front and back emission of 17 devices is 3.5% and 1.2%, respectively, and the histograms of the device EQE are shown in Supplementary Fig. 5. The transparent PeLED exhibited a short T$_{50}$ lifespan of 4 min (Supplementary Fig. 6), which is inferior to the lifespan shown in opaque PeLEDs[11]. This could be attributed to a slight plasma damage to the device that is inflicted by the ITO sputtering process.

In the pursuit of a transparent LED, sputtered ITO would appear to be a rather natural material candidate for the back electrode. However, we found that the thin polymer and perovskite active layers in a PeLED are particularly vulnerable to plasma damage from the ITO sputtering process, even when conducted at room temperature, unlike the thicker layers that were employed in semitransparent perovskite solar cells[12, 13]. Figure 2a shows the comparison of device characteristics between four representative PeLEDs that were respectively fabricated with Al (10 nm)/ITO (40 nm)/Ag (10 nm)/ITO (40 nm), ITO (500 nm), ITO (40 nm)/Ag (10 nm)/ITO (40 nm), and Al (80 nm) back electrodes. As shown in the current density vs. voltage plots, the 500-nm ITO PeLED has a 2–3 orders-of-magnitude higher current density before device turn-on (<1.5 V), compared to the Al/ITO/Ag/ITO and the Al PeLED devices. The electronic current in this low-voltage regime is generally dominated by leakage current and not by carrier injection into the semiconductor bands. Hence, this indicates that the ITO sputtering process has created a significant density of sub-gap defect states, most likely within the underlying poly-TPD and perovskite layer, which results in severe micro-shunting pathways and a large leakage current. The radiance achieved by the 500-nm ITO PeLED is also significantly lower overall driving voltages, since leakage currents do not contribute to radiative recombination and are essentially wasted as Joule heating.

We next employed an ITO (40 nm)/Ag (10 nm)/ITO (40 nm) sandwiched electrode structure[14], first to reduce the amount of ITO that we had to sputter in order to minimize damage, and also to provide a lower sheet resistance that is required for the device to operate efficiently. However, the ITO/Ag/ITO PeLED displayed similar high leakage currents and low radiance, which indicate that the sputtering of a thin ITO layer was still capable of causing damage when directly performed on top of the other active layers.

We have thus further implemented, in our transparent PeLED, a 10-nm Al interlayer to reduce direct plasma damage. The resulting Al/ITO/Ag/ITO electrode offers a significantly reduced leakage current and injects charges as efficiently as an opaque Al electrode, as shown by their identical turn-on voltage. We speculate that the Al interlayer offers more than a physical barrier, and that its conductive metallic nature has likely assisted in shielding the active layers from charge build-up and electrical breakdown during sputtering. However, it is still apparent that the Al/ITO/Ag/ITO PeLED possesses a higher leakage current compared to the Al counterpart, and thus performed less efficiently than the opaque Al PeLED. Nevertheless, the Al/ITO/Ag/ITO offers an optimal balance between low sheet resistance, high transparency, low damage, and efficient charge injection, and thus enabled the realization of a transparent PeLED with respectable optoelectronic performance.

Figure 2b shows the comparison of the optical transmittance between the Al/ITO/Ag/ITO and 500-nm ITO back electrodes, as well as that of their respective PeLEDs. The Al/ITO/Ag/ITO electrode and PeLED both show reasonably flat transmittance

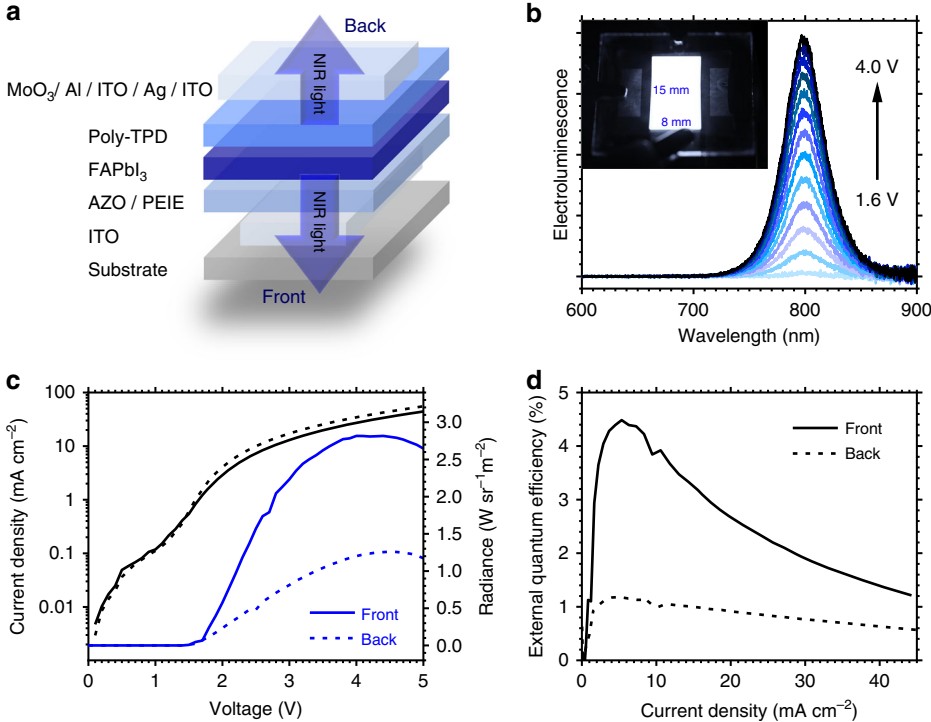

**Fig. 1 Architecture and performance of transparent perovskite light-emitting diode. a** Device structure of ITO/AZO/PEIE/FAPbI$_3$/Poly-TPD/MoO$_3$/Al/ITO/Ag/ITO PeLED. **b** Electroluminescence spectra of PeLED from 1.6 to 4.0 V. Inset shows the near-infrared photo of a 120-mm$^2$ PeLED. **c** Combined current density vs. voltage (black solid and dashed lines) and radiance vs. voltage plots (blue solid and dashed lines) of PeLED. **d** External quantum efficiency vs. current density plots of PeLED. Solid lines represent device measurements from the front and dashed lines represent measurements from the back.

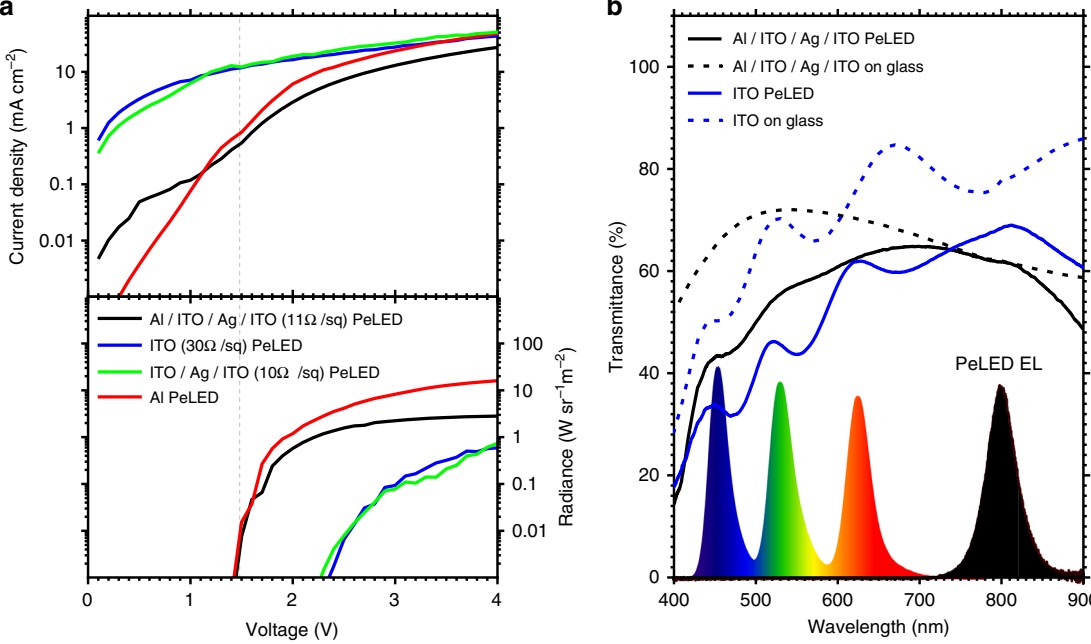

**Fig. 2 Electrical and optical characteristics of transparent perovskite light-emitting diodes. a** Comparison of current density (top graph) and radiance (bottom graph) between PeLEDs using Al/ITO/Ag/ITO (black line), ITO (blue line), ITO/Ag/ITO (green line), and Al (red line) as back electrodes. **b** Comparison of the optical transmittance of Al/ITO/Ag/ITO PeLED (black solid line) and ITO PeLED (blue solid line) and their respective Al/ITO/Ag/ITO (black dashed line) and ITO (blue dashed line) transparent electrodes on glass. Bottom plots show the emission spectra of a typical smart-watch display, and the NIR EL spectrum of the PeLED.

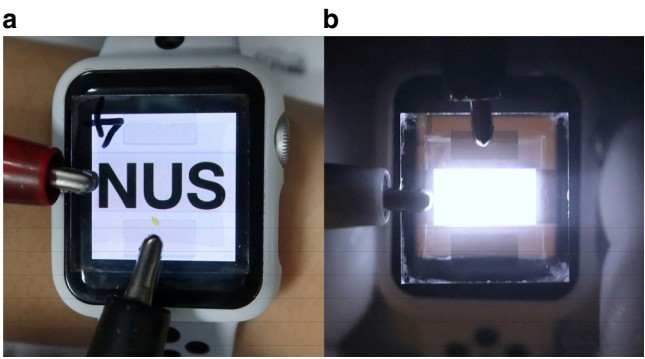

**Fig. 3 Demonstration of covert illumination function of a transparent perovskite light-emitting diode. a** A transparent PeLED overlaid across a smart-watch display to show high optical transparency and neutral color. **b** Near-infrared photo showing bright NIR electroluminescence from the transparent PeLED above the smart-watch display.

profiles across the visible and NIR region, compared to the 500-nm ITO electrode that absorbs more strongly in the blue spectral region. Notably, the Al/ITO/Ag/ITO PeLED possesses a high average transmittance of above 55% in the range of 450–650 nm, hence allowing them to be technologically relevant for electronic trichromatic display applications. The spectral profile of a typical smart-watch color display and that of our NIR PeLED is shown in Fig. 2b for reference.

**Transparent device applications**. We reasoned that the respectable NIR EL efficiency and the good optical transmittance of our PeLED will enable an exciting array of advanced covert illumination functions that was previously unachievable on small wearable gadgets. As a proof-of-concept, we overlaid our transparent PeLED over a smart-watch, as shown in Fig. 3a. A high-contrast, neutral-density white display can be observed across our PeLED, thus corroborating the above flat-spectral transmittance results. We operated the NIR PeLED at 3.2 V and imaged the smart-watch using a NIR camera, as shown in Fig. 3b. An intense NIR EL was observed, masking out the features on the underlying visible display. This demonstration shows that a transparent NIR LED could now be conceivably built onto a display to provide security and sensing functionalities, such as face recognition, eye-tracking, or motion and depth sensing, that were only recently possible on larger tablet computers and phones.

We have thus showcased, in this work, the successful fabrication of a transparent perovskite-based near-infrared light-emitting diode, and illustrated its functional application on a small wearable device. With the increasing consumer demand for advanced functionalities in these smart and highly connected devices, we anticipate that our concept could engender significant security functions and interactive sensing features in the next generation of technology gadgets, especially for devices that possess limited spatial real estate.

## Methods

**Perovskite precursor preparation**. The $FAPbI_3$ perovskite precursor was prepared by dissolving 27.7-mg FAI (Xi'an Polymer Light Technology), 33.2-mg $PbI_2$ (Sigma-Aldrich), and 7.7-mg 5-AVA (Sigma-Aldrich) in 1 ml of anhydrous N,N-dimethylformamide (Sigma-Aldrich). The precursor solution was stirred for 2 h at 80 °C in a nitrogen-filled glove box before use.

**Transparent PeLED fabrication**. Pre-patterned ITO-glass substrates (8 Ω/sq) were cleaned in detergent solution, deionized water, acetone, and isopropanol for 10 min, respectively, and then dried with a nitrogen gun. The substrates were treated in a UV–ozone cleaner for 15 min before spin-coating subsequent layers. AZO nanoparticles (2.5 wt% in IPA, Avantama) were deposited by spin-coating at 5000 rpm for 1 min, followed by annealing at 140 °C for 10 min. After cooling, a thin layer of

PEIE (0.4 wt% in 2-methoxyethanol) was spin-coated at 5000 rpm for 1 min. The layers were annealed at 110 °C for 20 min. The substrates were then transferred into a nitrogen-filled glove box for the deposition of subsequent layers. Eighty microliter of perovskite precursor solution was spin-coated at 3000 rpm for 1 min, followed by annealing at 100 °C for 16 min. This forms perovskite platelets with height of ~40 nm. The poly-TPD (American Dye Source) hole-transport layer was spin-coated from a chlorobenzene solution with a concentration of 13 mg ml$^{-1}$. $MoO_3$ (10 nm) and Al (10 nm) were sequentially thermal evaporated through a shadow mask at a pressure below $10^{-6}$ Torr. The substrates were then transferred into a sputtering system chamber (FHR) for ITO deposition. ITO (two layers, 40 nm each) were deposited at room temperature by pulsed DC magnetron sputtering from a cylindrical rotatable ceramic target ($In_2O_3$:$SnO_2$, 97:3 wt%), using a 205 sccm gas flow (Ar:$O_2$, 98:2) at a DC power of 2 kW. The Ag interlayer (10 nm), between the ITO layers, was deposited by thermal evaporation at a pressure below $10^{-5}$ Torr. The area of the device is 120 mm$^2$, and is defined by the overlap between the substrate ITO and the Al/ITO/Ag/ITO electrode.

**Transparent PeLED characterization**. The current density vs. voltage characteristics were measured using a Keithley 2450 source-measure unit. The voltage was swept from 0 to 5 V at 0.1 V steps and a delay time of 1 s. Simultaneously, the photon flux was measured using a 100 mm$^2$ Hamamatsu silicon photodiode with NIST traceable calibration at a distance of 110 mm. The EL spectra were recorded concurrently using an Ocean Optics Flame-T spectrometer. EQE was calculated by taking a Lambertian emission profile. The front and back emission from the same transparent device was measured separately in two current–voltage sweeps using the same configuration and settings. It is worth noting that the device may degrade slightly during the first current–voltage sweep. The base below the device is dark to minimize the collection of reflected light. All device measurements were performed in a dark enclosure in an argon-filled glove box. The NIR image of the PeLED was captured using an IR-modified Canon 200D DSLR camera.

**UV–visible–NIR transmittance spectroscopy**. The UV–visible–NIR transmittance spectra were obtained by measuring the transmitted light intensity of an Ocean Optics HL-2000 broadband light source with a calibrated Ocean Optics Flame-T spectrometer. The transmittance of the electrodes on glass was measured using an Agilent CARY-7000 spectrophotometer.

## Data availability
The datasets generated during and/or analyzed during the current study are available from the corresponding author on reasonable request.

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

## Acknowledgements

We are grateful to Prof. Armin Aberle for providing equipment and resources to support this work. We acknowledge Dr Selvaraj Venkataraj for his assistance with the operation of the ITO sputtering equipment. The authors are grateful for funding support from the Ministry of Education of Singapore (R-143-000-674-114 and R-143-000-691-114) and the National University of Singapore (R-143-000-639-133 and R-143-000-A10-133).

## Author contributions

C.X. fabricated the devices and performed the measurements. X.Z. and E.W.Y.O. assisted with device fabrication, characterization, and analysis. Z.-K.T. and C.X. analyzed the data and wrote the paper. Z.-K.T. initiated the project and guided the work.

## Competing interests

The authors declare no competing interests.
