## [Peer Review File · Nature Communications]

REVIEWER COMMENTS

Reviewer #1 (Remarks to the Author):

In this manuscript, Xie et al. fabricate a large-area, transparent NIR perovskite LED and demonstrate its potential applications as illumination integrated in a color display. This provides a new strategy for the light source in large-area, flexible wearable devices. In general, this is a very interesting work. I would recommend acceptance of this manuscript after minor revision.

1. I am curious why the transparent device has a low EQE of 5.7% with device area of 120 mm². In their previous work (Nature Photonics, 14, 215-218 (2020)), they already demonstrate 12.1% EQE of 900 mm² LEDs with conventional architecture. Is this due to the absorption of top electrode or the degradation of device during the multi-scans?
2. The authors should provide the statistic of device efficiency.
3. Could the authors use other transparent electrodes to replace the Al/ITO/Ag/ITO? The 55% transmittance is not good enough for transparent LED application.
4. It might be helpful if the authors could provide the optimization process of the top electrode.

Reviewer #2 (Remarks to the Author):

This work is very interesting and would potentially meet the standards for publication on Nature Communications. However, some revisions are needed before acceptance.

In this manuscript the Authors reported a transparent perovskite light-emitting diode (LED) that could be overlaid across an emissive colour display to provide efficient and high-intensity covert NIR illumination. The devices presented here are constructed with an ITO/AZO/PEIE/FAPbI₃/poly-TPD/MoO₃/Al/ITO/Ag/ITO architecture, and offer a high average transmittance of more than 55% across the visible spectral region. In particular, the Al/ITO/Ag/ITO top transparent electrode was designed to offer a combination low sheet resistance and low plasma damage during device fabrication. The devices emit at 799 nm with a high total external quantum efficiency of 5.7% at a current density of 5.3 mA cm⁻² and a high radiance of 1.5 W sr⁻¹ m⁻², and possess a large functional device area of 120 mm². This report represents an advancement of the perovskite-based light emitting devices as it could be integrated in battery-powered wearable devices, and could enable new security and remote sensing features on future smart-watches, phones and augmented or virtual reality headsets. The approach is then very interesting and novel to suit the standards for publication on Nature Communications, although the work requires some clarifications/modification to finally express the research potential.

Specific comments

- The authors shown and discussed how the Al/ITO back electrode perform better than the pristine sputtered ITO, further reasoning about a process (sputtering) driven degradation of the perovskite layer. However alternative effects could take place in justifying the observation, particularly due to the reflective nature of the Al layer potentially leading to cavity effects. Could the authors deepen their claim? We suggest adding some structural (XRD) and optical measurements (PL and time resolved PL) to show how the ITO sputtering is really reducing the quality of the perovskite film.
- Related to this point we believe the characterization of the perovskite material is somehow missing in this work, and we are sure the addition of data in this regard could beneficially affect the impact and the soundness of the work.
- We are wondering if the authors tested the structural nature and stability of the FAPbI₃ perovskite, it is well known the polymorphism of this compound and the need of using elevated annealing temperature to obtain the optoelectronic active alpha phase. Since the authors used only 100 °C

annealing temperature they should also show and discuss some XRD results.

- What is the thickness of the perovskite active layer in those devices? Did the authors screen the devices performances of differently thick active layers?
- Did the Authors test the stability of their devices?

Reviewer #3 (Remarks to the Author):

The authors report on the fabrication of semi-transparent NIR emitting perovskite LEDs. Although I find the work interesting, I believe that it cannot be accepted for publication in Nature Communications. Following the issues:

- 1) The work looks a mere exercise of device optimization;
- 2) The literature context is not well explained;
- 3) The English need a revision.

Response to Reviewers

Reviewer #1 (Remarks to the Author):

In this manuscript, Xie et al. fabricate a large-area, transparent NIR perovskite LED and demonstrate its potential applications as illumination integrated in a color display. This provides a new strategy for the light source in large-area, flexible wearable devices. In general, this is a very interesting work. I would recommend acceptance of this manuscript after minor revision.

We are grateful for the positive and constructive comments from the reviewer. We have addressed the comments and revised the manuscript accordingly.

1. I am curious why the transparent device has a low EQE of 5.7% with device area of 120 mm². In their previous work (Nature Photonics, 14, 215-218 (2020)), they already demonstrate 12.1% EQE of 900 mm² LEDs with conventional architecture. Is this due to the absorption of top electrode or the degradation of device during the multi-scans?

The sputtering of ITO causes plasma damage to the device active layers, primarily the underlying poly-TPD and perovskite layer, and thus causes the transparent PeLED to perform less efficiently. We have added the detailed discussion below in page 4 of our manuscript to explain the observed device performance.

Figure 2a shows the comparison of device characteristics between four representative PeLEDs that were respectively fabricated with Al(10 nm)/ITO(40 nm)/Ag(10 nm)/ITO(40 nm), ITO(500 nm), ITO(40 nm)/Ag(10 nm)/ITO(40 nm) and Al(80 nm) back electrodes. As shown in the current density vs. voltage plots, the 500 nm ITO PeLED has a 2-3 orders-of-magnitude higher current density before device turn-on (<1.5 V), compared to the Al/ITO/Ag/ITO and the Al PeLED devices. The electronic current in this low voltage regime is generally dominated by leakage current and not by carrier injection into the semiconductor bands. Hence, this indicates that the ITO sputtering process has created a significant density of sub-gap defect states, most likely within the underlying poly-TPD and perovskite layer, which results in severe micro-shunting pathways and a large leakage current. The radiance achieved by the 500 nm ITO PeLED is also significantly lower over all driving voltages, since leakage currents do not contribute to radiative recombination and are essentially wasted as Joule heating.

We next employed an ITO(40 nm)/Ag(10 nm)/ITO(40 nm) sandwiched electrode structure,¹⁴ first to reduce the amount of ITO that we had to sputter in order to minimize damage, and also to provide a lower sheet resistance that is required for the device to operate efficiently. However, the ITO/Ag/ITO PeLED displayed similar high leakage currents and low radiance, which indicate that the sputtering of a thin ITO layer was still capable of causing damage when directly performed on top of the other active layers.

We have thus further implemented, in our transparent PeLED, a 10 nm Al interlayer to reduce direct plasma damage. The resulting Al/ITO/Ag/ITO electrode offers a significantly reduced leakage current and injects charges as efficiently as an opaque Al electrode, as shown by their identical turn-on voltage. We speculate that the Al interlayer offers more than a physical barrier, and that its conductive metallic nature has likely assisted in shielding the active layers from charge build-up and electrical breakdown during sputtering. However, it is still apparent that the

Al/ITO/Ag/ITO PeLED possesses a higher leakage current compared to the Al counterpart, and thus performed less efficiently than the opaque Al PeLED. Nevertheless, the Al/ITO/Ag/ITO offers an optimal balance between low sheet resistance, high transparency, low damage and efficient charge injection, and thus enabled the realisation of a transparent PeLED with respectable optoelectronic performance.

2. The authors should provide the statistic of device efficiency.

The average max-EQE for the front and back emission of 17 devices are 3.5% and 1.2%, respectively, and the histograms of the device EQE are shown in Supplementary Figure S5.

3. Could the authors use other transparent electrodes to replace the Al/ITO/Ag/ITO? The 55% transmittance is not good enough for transparent LED application.

The use of other transparent electrodes is described in our detailed response to comment 1 above. Unfortunately, we have not attained an electrode with above 55% transmittance that possess good performance despite notable efforts. We hope to improve this in our future work.

4. It might be helpful if the authors could provide the optimization process of the top electrode.

We have described our electrode optimization process in our detailed response to comment 1 above.

Reviewer #2 (Remarks to the Author):

This work is very interesting and would potentially meet the standards for publication on Nature Communications. However, some revisions are needed before acceptance.

In this manuscript the Authors reported a transparent perovskite light-emitting diode (LED) that could be overlaid across an emissive colour display to provide efficient and high-intensity covert NIR illumination. The devices presented here are constructed with an ITO/AZO/PEIE/FAPbI₃/poly-TPD/MoO₃/Al/ITO/Ag/ITO architecture, and offer a high average transmittance of more than 55% across the visible spectral region. In particular, the Al/ITO/Ag/ITO top transparent electrode was designed to offer a combination low sheet resistance and low plasma damage during device fabrication. The devices emit at 799 nm with a high total external quantum efficiency of 5.7% at a current density of 5.3 mA cm⁻² and a high radiance of 1.5 W sr⁻¹ m⁻², and possess a large functional device area of 120 mm². This report represents an advancement of the perovskite-based light emitting devices as it could be integrated in battery-powered wearable devices, and could enable new security and remote sensing features on future smart-watches, phones and augmented or virtual reality headsets. The approach is then very interesting and novel to suits the standards for publication on Nature communications, although the work requires some clarifications/modification to finally express the research potential.

We are grateful for the positive and constructive comments from the reviewer. We have addressed the comments and revised the manuscript accordingly.

Specific comments

- The authors shown and discussed how the Al/ITO back electrode perform better than the pristine sputtered ITO, further reasoning about a process (sputtering) driven degradation of the perovskite layer. However alternative effects could take place in justifying the observation, particularly due to the reflective nature of the Al layer potentially leading to cavity effects. Could the authors deepen their claim? We suggest adding some structural (XRD) and optical measurements (PL and time resolved PL) to show how the ITO sputtering is really reducing the quality of the perovskite film.

The damage caused by the ITO sputtering is best characterized through electrical studies, as we elaborate in detail below. It is difficult to determine such damage spectroscopically because the optical cross-section of such damage is remarkably small and the presence of multiple optically-active layers in a device further complicates the analysis. The presence of electrodes in a device also quenches PL and thus makes it difficult to assess film quality using quantitative PL methods.

We have included the detailed discussion below in page 4 of the manuscript.

Figure 2a shows the comparison of device characteristics between four representative PeLEDs that were respectively fabricated with Al(10 nm)/ITO(40 nm)/Ag(10 nm)/ITO(40 nm), ITO(500 nm), ITO(40 nm)/Ag(10 nm)/ITO(40 nm) and Al(80 nm) back electrodes. As shown in the current density vs. voltage plots, the 500 nm ITO PeLED has a 2-3 orders-of-magnitude higher current density before device turn-on (<1.5 V), compared to the Al/ITO/Ag/ITO and the Al PeLED devices. The electronic current in this low voltage regime is generally dominated by leakage current and not by carrier injection into the semiconductor bands. Hence, this indicates that the ITO sputtering process has created a significant density of sub-gap defect states, most likely within the underlying

poly-TPD and perovskite layer, which results in severe micro-shunting pathways and a large leakage current. The radiance achieved by the 500 nm ITO PeLED is also significantly lower over all driving voltages, since leakage currents do not contribute to radiative recombination and are essentially wasted as Joule heating.

We next employed an ITO(40 nm)/Ag(10 nm)/ITO(40 nm) sandwiched electrode structure,¹⁴ first to reduce the amount of ITO that we had to sputter in order to minimize damage, and also to provide a lower sheet resistance that is required for the device to operate efficiently. However, the ITO/Ag/ITO PeLED displayed similar high leakage currents and low radiance, which indicate that the sputtering of a thin ITO layer was still capable of causing damage when directly performed on top of the other active layers.

We have thus further implemented, in our transparent PeLED, a 10 nm Al interlayer to reduce direct plasma damage. The resulting Al/ITO/Ag/ITO electrode offers a significantly reduced leakage current and injects charges as efficiently as an opaque Al electrode, as shown by their identical turn-on voltage. We speculate that the Al interlayer offers more than a physical barrier, and that its conductive metallic nature has likely assisted in shielding the active layers from charge build-up and electrical breakdown during sputtering. However, it is still apparent that the Al/ITO/Ag/ITO PeLED possesses a higher leakage current compared to the Al counterpart, and thus performed less efficiently than the opaque Al PeLED. Nevertheless, the Al/ITO/Ag/ITO offers an optimal balance between low sheet resistance, high transparency, low damage and efficient charge injection, and thus enabled the realisation of a transparent PeLED with respectable optoelectronic performance.

- Related to this point we believe the characterization of the perovskite material is somehow missing in this work, and we are sure the addition of data in this regard could beneficially affect the impact and the soundness of the work.

To ensure completeness to this work, we have now included the SEM, XRD and PL characteristics of the perovskite layer in Supplementary Figure S1, S2 and S3, respectively. We have also updated the main text to include these measurements.

- We are wondering if the authors tested the structural nature and stability of the FAPbI₃ perovskite, it is well known the polymorphism of this compound and the need of using elevated annealing temperature to obtain the optoelectronic active alpha phase. Since the authors used only 100 °C annealing temperature they should also show and discuss some XRD results.

We performed XRD measurements on the perovskite layer and found it to remain stable after 3 months of storage in an argon environment. The results are shown in Supplementary Figure S2 above.

- What is the thickness of the perovskite active layer in those devices? Did the authors screen the devices performances of differently thick active layers?

The perovskite layer comprises platelets with height of approximately 40 nm. We have optimised the perovskite layer in our previous work (Zhao, X. & Tan, Z.-K. *Nature Photonics* 2020, 14, 215) and have not done the same in this study, since the present work is primarily focused on the transparent device structure.

We have updated the thickness of the perovskite in 'Methods'.

- Did the Authors test the stability of their devices?

The transparent PeLED exhibited a short T₅₀ lifespan of 4 minutes (Supplementary Figure S6), which is inferior to the lifespan shown in opaque PeLEDs. This could be attributed to a slight plasma damage to the device that is inflicted by the ITO sputtering process.

We have included this discussion on page 3 of the manuscript.

REVIEWERS' COMMENTS:

Reviewer #1 (Remarks to the Author):

The authors have revised their manuscript with additional experiments that address my concerns. I'm happy to recommend the manuscript for acceptance in Nature Communications.

Reviewer #2 (Remarks to the Author):

The manuscript in its revised version well takes into account the reviewer's suggestions, resulting improved if compared to the previous submission. For these reasons it would suit the standard for publication on Nature Communications.